# Systematic review of animal-based indicators to measure thermal, social, and immune-related stress in pigs

Raúl David Guevara[1,2,3]*, Jose J. Pastor[4], Xavier Manteca[1,2], Gemma Tedo[4], Pol Llonch[1,2]

1 Department of Animal and Food Science, Universitat Autònoma de Barcelona, Barcelona, Bellaterra (UAB), Spain, 2 Animal Welfare and Nutrition Service (SNiBA), Department of Animal and Food Science, Universitat Autònoma de Barcelona, Cerdanyola del Vallès, Spain, 3 Advisors S.L., Research Park UAB, Campus UAB, Cerdanyola del Vallès, Spain, 4 Innovation Division, Lucta S.A., UAB Research Park, Campus UAB, Cerdanyola del Vallès, Spain

* raul.guevara@awec.es

**Data Availability Statement:** All relevant data are within the paper and its Supporting Information files.

**Funding:** The author(s) received no specific funding for this work.

## Abstract

The intense nature of pig production has increased the animals' exposure to stressful conditions, which may be detrimental to their welfare and productivity. Some of the most common sources of stress in pigs are extreme thermal conditions (thermal stress), density and mixing during housing (social stress), or exposure to pathogens and other microorganisms that may challenge their immune system (immune-related stress). The stress response can be monitored based on the animals' coping mechanisms, as a result of specific environmental, social, and health conditions. These animal-based indicators may support decision making to maintain animal welfare and productivity. The present study aimed to systematically review animal-based indicators of social, thermal, and immune-related stresses in farmed pigs, and the methods used to monitor them. Peer-reviewed scientific literature related to pig production was collected using three online search engines: ScienceDirect, Scopus, and PubMed. The manuscripts selected were grouped based on the indicators measured during the study. According to our results, body temperature measured with a rectal thermometer was the most commonly utilized method for the evaluation of thermal stress in pigs (87.62%), as described in 144 studies. Of the 197 studies that evaluated social stress, aggressive behavior was the most frequently-used indicator (81.81%). Of the 535 publications examined regarding immune-related stress, cytokine concentration in blood samples was the most widely used indicator (80.1%). Information about the methods used to measure animal-based indicators is discussed in terms of validity, reliability, and feasibility. Additionally, the introduction and wide spreading of alternative, less invasive methods with which to measure animal-based indicators, such as cortisol in saliva, skin temperature and respiratory rate via infrared thermography, and various animal welfare threats via vocalization analysis are highlighted. The information reviewed was used to discuss the feasible and most reliable methods with which to monitor the impact of relevant stressors commonly presented by intense production systems on the welfare of farmed pigs.

**Competing interests:** The authors have declared that no competing interests exist.

## Introduction

Pig production has evolved towards highly intense, large-scale production systems, with an associated increase in the animals' exposure to stressors that can affect their welfare and production efficiency [1].

Stressors are any environmental, physiological, or social factor that causes poor animal welfare [2]. Pigs are subjected to stressors, such as extreme temperatures [3], which generate thermal stress [4–10], high density and mixed housing, which causes social stress [11–17], and crowded spaces, which might increase the transmission of pathogens and other microorganisms that may challenge the immune system, causing immune-related stress [18–22].

The stress response is a physiological response aimed at maintaining the body's physiological balance when an individual is experiencing effects of a stressor [23, 24]. However, this response is associated with a reduction in productive efficiency due to several mechanisms, such as a reduction in appetite and feed intake, an increase in energy and nutrient expenditure, and a higher susceptibility to infectious diseases [25]. This reduction in efficiency in turn directly impacts economic profitability. Furthermore, the negative impact on animal welfare from the exposure to stressors has a negative impact on consumer perceptions of animal products and reduces their acceptability of these products [26].

The use of animal-based indicators to evaluate animals' responses to specific circumstances is a general practice in current animal production systems (i.e., feed quality, environmental conditions, physiological development, and social interactions, among others) [25]. Animal-based indicators are obtained accurately and are usually presented quantitatively (measurement method). Of note, there is a difference between the indicator concept and the method concept, due to which the measurement method can limit the intention of the information interpreted as an indicator [26]. These indicators (usually a set of indicators, which provides better conclusions) allow the evaluation of the animals' efficiency in the use of resources and predict animal performance under certain conditions. Additionally, animal-based indicators should fulfil the concepts of validity (the fitness of an indicator = properly developed, optimized, and standardized for an intended purpose), reliability (the ability of an indicator to be used under different conditions by different persons while still producing similar results), and feasibility (the practical application and use of an indicator under different circumstances) to generate accurate and trustworthy observations [26, 27].

Over the last 20 years, several indicators have been proposed for use in monitoring the status of pigs. Physiological markers [28–36], performance and body measurements [37–42], and behavioral parameters [43–53] are some examples of animal-based indicators used to assess the growth and performance of animals during production. The detection and measurement of several indicators, however, involve some degree of invasiveness, which can cause discomfort, stress, and fear reactions, altering the comfort and behavior of the animals and likely the accuracy of the measurements [54]. Therefore, it is important to invest resources in the development of methods and techniques that minimize animal stress during measurements in an effort to improve both the quality of the measurements and the welfare of the animal. The optimization and development of less invasive or less stressful measurement techniques would fulfill the *refinement* concept from the "*three R's*" principle of animal research (*replacement*, *reduction*, *refinement*) defined by Russell [55]. Knowing a variety of animal-based indicators and features of the measurement techniques will provide researchers with the tools with which to make decisions regarding the techniques available to assess animal performance and welfare, therefore easing experimental design and practical execution.

The aim of the present study was to review animal-based indicators described in current literature to monitor the impact of different sources of stress in pigs and to highlight new

methods and techniques that may refine the monitoring of individual pigs under intense farming conditions. Therefore, a systematic literature review was performed to identify, classify, and discuss the different methods used in peer-reviewed literature, in order to detect physiological, behavioral, and performance information from pigs under the stress provoked by heat, social, and immunological challenges.

## Materials and methods

### Search criteria, strategy, and study selection criteria

Following the Preferred Reporting Items for Systematic Reviews and Meta-Analyses (PRISMA) guidelines [56], a single researcher (RDG) performed a literature search, which was verified by two other researchers (GT and PL). The papers found during this systematic search were verified via discussions among the researchers (RDG, GT, and PL) regarding the feasibility of the methods mentioned in the manuscripts collected. A systematic review was performed to identify animal-based indicators of the following three key sources of stress found in pig production: 1. thermal stress; 2. social stress; and 3. immune-related stress. Peer-reviewed scientific literature related to pig production published from 2000 to 2020 was collected from three online search engines: ScienceDirect and Scopus from Elsevier, and PubMed from the National Center for Biotechnology Information (United States of America). An independent systematic search was performed for each stress model. Manuscripts were reviewed if they had at least one open-access abstract, and articles in which the abstract was the only open-access section were counted as valid if information about animal-based indicators and measurement techniques were explicitly mentioned in the available text. Studies performed on species other than pigs, *in vitro*, reproductive, transportation-related, and social isolation studies, as well as manuscripts in languages other than English, were omitted, as well as literature reviews, due to the absence of details about the techniques used for the measurement of indicators. The included studies were all performed under research or production conditions.

The search terms to find relevant literature were selected based on the primary research question: "*what are the animal-based indicators with which to measure the impact of stress from different sources (thermal, social, and immune-related) in pigs*?" The search criteria were divided into three main categories: 1. the stress model; 2. the animal model; and 3. the possible indicators (invasive, less invasive, and non-invasive) used to detect the impacts of stress on the animals. Therefore, the query line was generated using a combination of three search terms (one from each search term category) within the *title*, *abstract*, and *keyword* sections (Table 1).

Valid papers were processed to extract information, such as animal-based indicators used to measure the impact of the stressor, the method utilized to measure that indicator, and the animals used in the study (including sample size, age, and breed). Additionally, the

**Table 1. Search terms used in the systematic searches for thermal, social, and immune-related stresses.**

| Animal | Thermal stress | | Social stress | | Immune-related stress | |
|---|---|---|---|---|---|---|
| | **Stress** | **Indicator** | **Stress** | **Indicator** | **Stress** | **Indicator** |
| Pigs | Heat stress | Physiology | Social stress | Cortisol | Immunological stress | Cortisol |
| Swine | Heat load | Lying behavior | Biting | Testosterone | Immunological challenge | Hormones |
| Piglets | Temperature humidity index | Body temperature | Tail biting | Flight behavior | Immune challenge | Cytokine |
| Porcine | Hyperthermia | Respiration rate | Contest | Fight behavior | Pathogen infection | Body temperature |
| Swine welfare | Thermal stress | Skin temperature | Agonistic behavior | Skin lesions | Immune response | Interleukin |
| Sow | Oxidative stress | Vocalizations | Beating | | Inflammatory response | Animal performance |
| | Respiratory alkalosis | Vitamin E | | | Lipopolysaccharide | Glucocorticoids |

immunological challenges used in the immune-related stress model were recorded. Due to the nature of the data extracted from the articles (animal-based indicators and measurement methods), only studies with a bioethical committee approval were considered for the review, in order to reduce the risk of bias, as measurement protocols should be previously approved for the performance of the study. No mathematically computed data were extracted for this review, as the data extracted were qualitative. The primary objective was to quantify the animal-based indicators and measurement techniques utilized in the available literature. Data extraction and bias assessment were performed by a single researcher (RDG), and validation was performed by two researchers (GT and PL).

Processed papers were classified based on the animal-based indicators used to assess the impact of each stressor, as well as the method used to measure each indicator. Each paper could fit into multiple categories, as most studies measured several animal-based indicators.

The goal of including the term "vitamin E" was to find studies that measure this metabolite as an indicator of the oxidative stress level in the animals under thermal stress conditions. Vitamin E is the primary antioxidant cell protector [57], and can be used as an antioxidant supplement in porcine diets to neutralize free radicals and reduce oxidative damage resulting from thermal stress [30, 58, 59].

## Results

### Thermal stress

The systematic search was conducted until December 1, 2020, and articles selected included only manuscripts in which the authors assessed the animals' thermal status under either production or research conditions (including any kind of experimental treatment). The systematic search for thermal stress articles yielded 3,239 results between the 3 search engines. After removing duplicates, all papers were reviewed and those that were not related to the search topic were excluded, leaving a total of 166 articles. After the final round of refinement, which included an inspection of the title, abstract, and materials and methods, and the removal of articles wrongly accepted during the first selection step because they were performed in other animal species, such as guinea pigs, or were performed without a real measurement of the thermal status of the animals, there was a total of 144 manuscripts (Fig 1). A list of references found in the systematic search is available in the S1 Table in S1 File.

Of the 144 articles obtained from the systematic review, there were 39 methods identified (Fig 2). All of the methods found in the literature, as well as the number of articles that used them, are presented in the S2 Table in S1 File. These methods were used to measure the thermal status of the animals via the detection of specific animal-based indicators.

The papers were classified into 7 indicator categories: body temperature (97 articles, 67.83% of the articles from the thermal stress search results); respiratory rate (76 articles, 53.14%); physiological markers (68 articles, 47.55%); skin temperature (58 articles, 40.55%); environmental indices (34 articles, 23.77%); behavior (25 articles, 17.48%); and animal vocalizations (8 articles, 5.59%). Physiological markers include all physiological variables reported in the literature related to blood chemistry, organ integrity assessment, and stress responses. While behavior focused on lying, assuming that pigs under thermal spent increased time lying down to facilitate heat dissipation [47, 60, 61]. Animal vocalizations were considered an individual indicator category, separate from behavior, because the publications reviewed studied the meaning of the sounds produced by the animals, and the researchers tried to relate the characteristics of the sounds with the animals' environmental, social, or health situations [62].

The inclusion of environmental indices (environmental-based indicators) is justified, as the severity of the thermal stress model can be estimated from environmental information. For

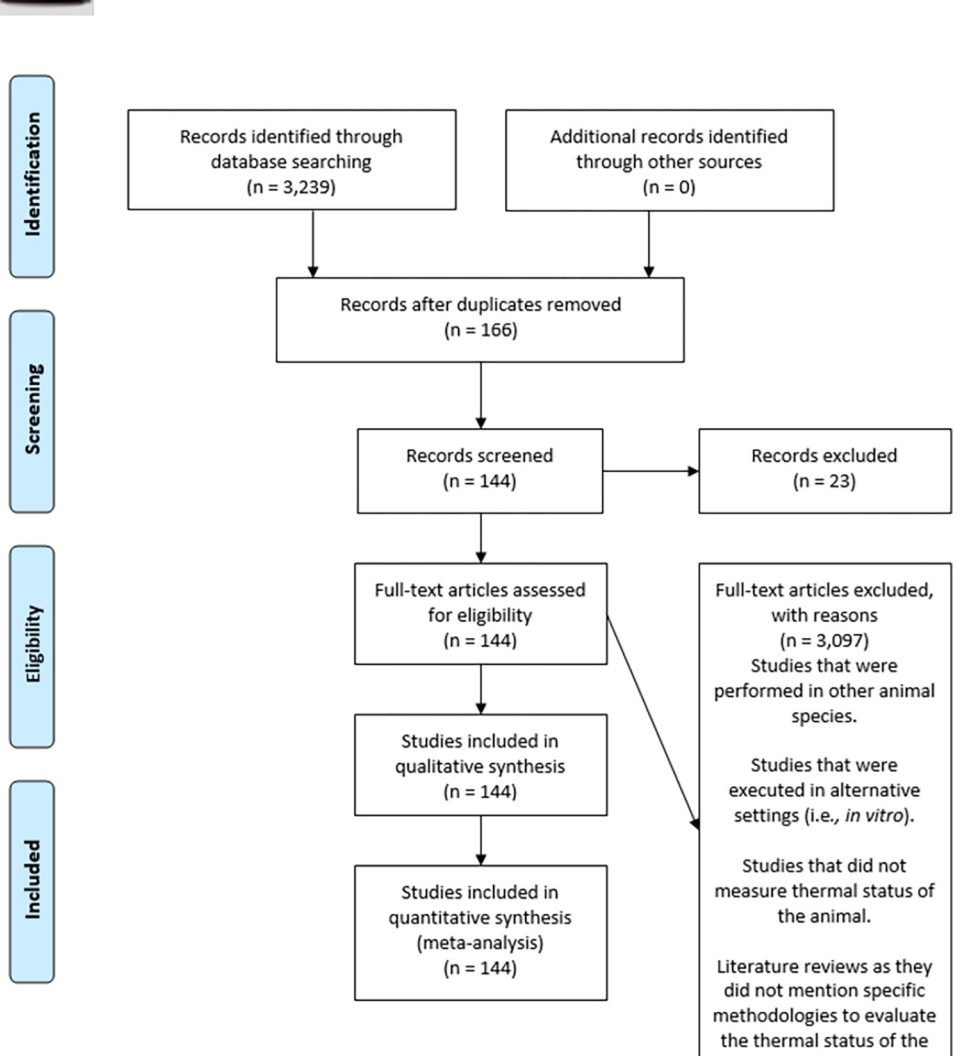

**Fig 1. PRISMA flow diagram for thermal stress systematic search results.**

example, the temperature and humidity index (THI), calculated using environmental variables, has been used in several studies to indirectly measure if pigs are being subjected to thermal stress [44, 57, 63–65]. The environmental variables are as follows: temperature (T) and relative humidity (RH). THI is calculated using the equation defined by the National Research Council (NRC) in 1971 [66], where T is the maximum daily temperature in degrees Celsius, and RH is the minimum daily humidity ranging from 0 to 100 [44, 63, 64, 66].

$$THI = (1.8 \times T + 32) - [0.55 - (0.0055 \times RH)] \times (1.8 \times T - 26)$$

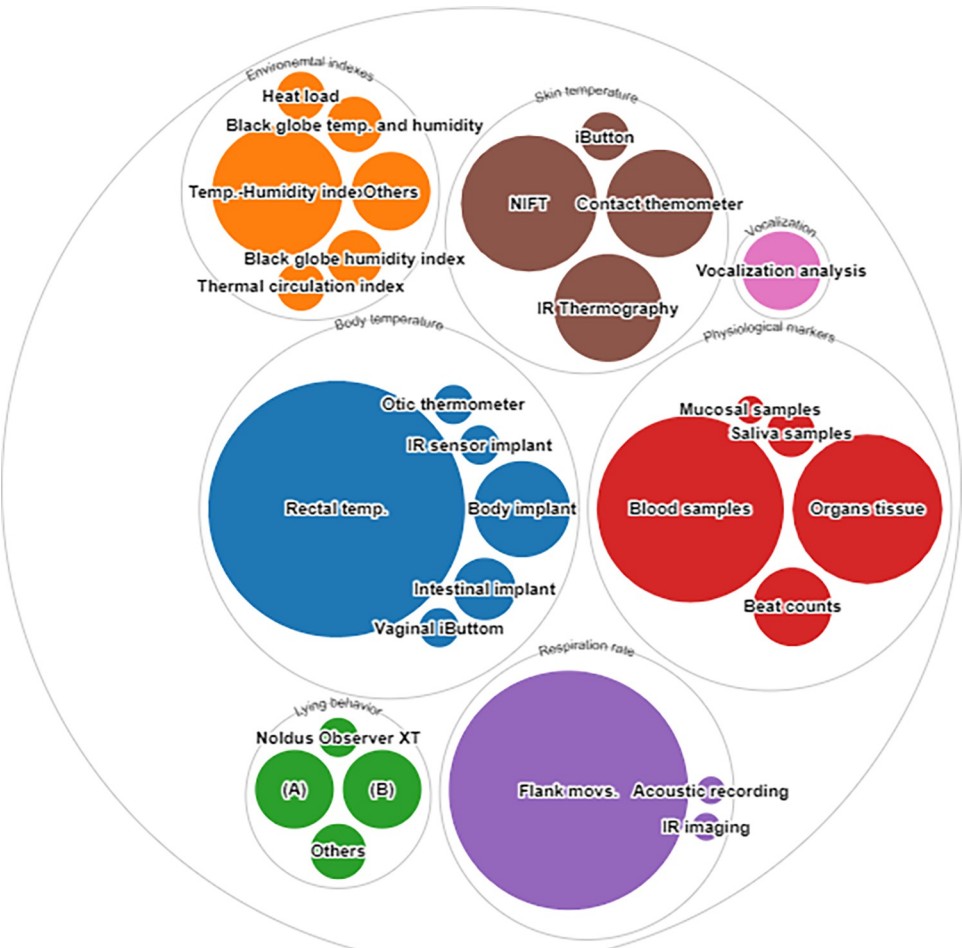

**Fig 2. Methods used to detect thermal stress indicators from the selected literature: (A) scan sampling; (B) video behavior classification.** Methods pooled in the "Others" domain are presented in the S2 Table in S1 File.
Figure generated through rawgraphs.io.

Of the physiological markers described, 33 biological parameters were identified that were significantly altered by the heat stress challenge (Fig 3).

Rectal temperature was the most frequently used method for measuring temperature, used as an indicator of thermal stress, as mentioned in 85 manuscripts (87.62% of the articles that measured body temperature).

Assessment of respiratory rate was the second most frequent indicator of thermal stress (76 articles, 53.14%). The frequency of flank movements was the primary method used to measure respiration rate (74 out of 76 articles).

The primary physiological indicators of stress were an increase in glucose, reduction in ileum integrity, and increase in cortisol concentration. Of these indicators, glucose concentration (20 articles, 29.11%) was the most frequently used biological marker. Additionally, blood sampling was the most commonly used method to detect changes in proteomics triggered by thermal stress (45 articles, 66.1%).

Environmental indices were used in 34 articles. The primary environmental index used to measure the impact of environmental temperature on animal physiology was the temperature-humidity index (22 articles, 64.7%). Other environmental indices used in the literature found

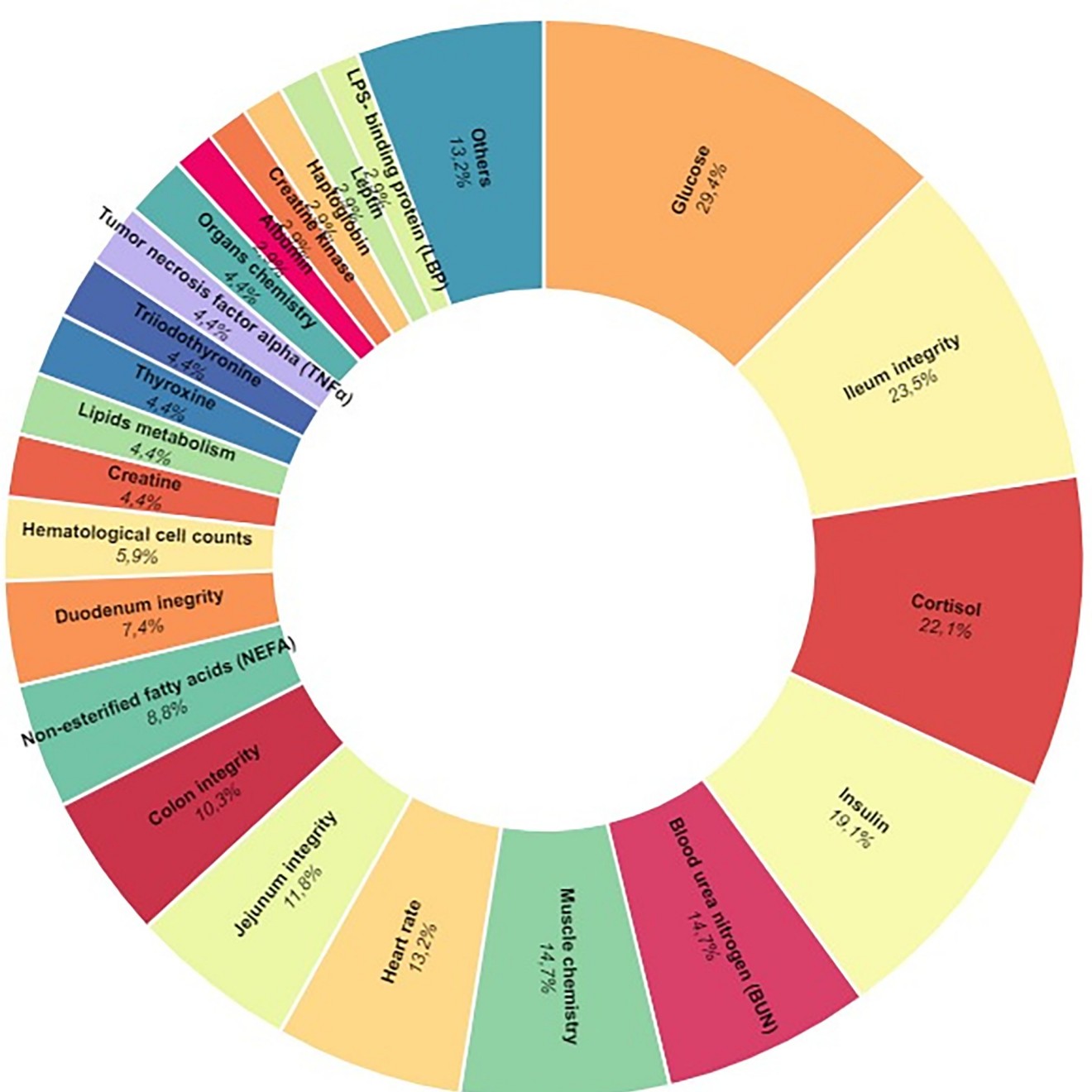

**Fig 3. Physiological markers used to evaluate thermal stress.** Other markers include: amino acids concentration; creatinine; gene expression; glutathione; lactate; myeloperoxidase; methylhistidine; and vitamin E. The comprehensive list of physiological markers found in the literature and the number of articles that used them are presented in the S3 Table in S1 File. Figure generated through rawgraphs.io.

were the calculation of the head load and the thermal circulation index. Non-contact infrared thermography (NIFT; 24 articles, 41.37%) was the technique most applied to measure the skin temperature of the pigs among the literature reviewed, other methods used were the contact thermometer and the thermographic camera.

For the assessment of behavior, scan sampling and video behavior classification (a trained observer counted the frequency of certain behaviors defined by an ethogram) were the most commonly used methods, reported in 8 articles (32%) each.

Overall, the analysis of vocalizations (8 articles, 5.59%) was the indicator category least reported in the literature about thermal stress.

## Social stress

The systematic search for studies regarding social stress was completed on December 2, 2020. In the studies selected for this group, social mixing occurred at some point during the rearing process. Isolation and new environmental test studies were excluded from the list. A total of 2,409 results were obtained, of which 1,431 were related to the search topic. After excluding duplicates, 229 references were used for the literature review. The final evaluation based on title, abstract, and materials and methods allowed us to exclude an additional 31 papers because these studies were either performed in other animal species, such as guinea pigs, or were performed in other settings, such as *in vitro*, genetic, reproductive, or transportation studies. In total, 197 references were used for the literature review and indicator summary (Fig 4). A list of manuscripts found in the systematic literature search is presented in the S4 Table in S1 File.

From the manuscripts found involving the social stress model, a total of 49 assessment methods were identified (Fig 5). Physiology-related measurement techniques evaluated 47 physiological markers, of which 2 techniques were utilized to evaluate skin lesions in the animals (either body or tail lesions), and 8 behaviors were observed through the evaluation of social interactions (Fig 6). A list of the methods and indicators is presented in the S5 Table in S1 File. The studies reviewed described methods utilizing four animal-based indicators to evaluate the effects of the social interactions of pigs.

Manuscripts were classified into 5 categories using animal-based indicators: social behavior (162 manuscripts, 81.81% of the literature reviewed regarding the social stress model), body lesions (118 manuscripts, 59.59%), animal performance (54 manuscripts, 27.27%), physiological markers (133 manuscripts, 67.17%), and vocalizations (7 manuscripts, 3.53%). Physiological markers utilized included all of the physiological variables reported in the literature related to blood chemistry, organ integrity assessment, and heart rate, among others. Social interaction measurements focused on agonistic behaviors among animals, such as aggression, intimidation, fights, etc.

The observation of social interactions was the most widely used indicator in the literature reviewed which quantified social interactions (162 manuscripts, 58.02% of the relevant literature). Direct observation was the most common method with which these interactions were detected.

Lesion assessment was performed separately for body (front, middle, and rear body areas, excluding the tail) and tail lesions. Overall, body lesion evaluation was the most commonly used method (97.45%), compared with tail lesion evaluation (19.49%). Additionally, the lesion score was the most widely used method to evaluate this indicator category (64 manuscripts, 54.23%) relative to lesion count.

Of the manuscripts reviewed, 133 (67.17%) articles identified 47 physiological markers, the most common of which were the detection of increases in the concentration of cortisol (84 manuscripts, 63.15%), lactate (7 manuscripts, 5.26%), glucose (6 manuscripts, 4.51%), catecholamines (6 manuscripts, 4.51%), adrenocorticotropic hormone (ACTH; 5 manuscripts, 3.75%), and haptoglobin (5 manuscripts, 3.75%). Blood sampling was the most widely used method described in manuscripts involving biochemical indicators (57.14%), followed by saliva sampling (45.11%).

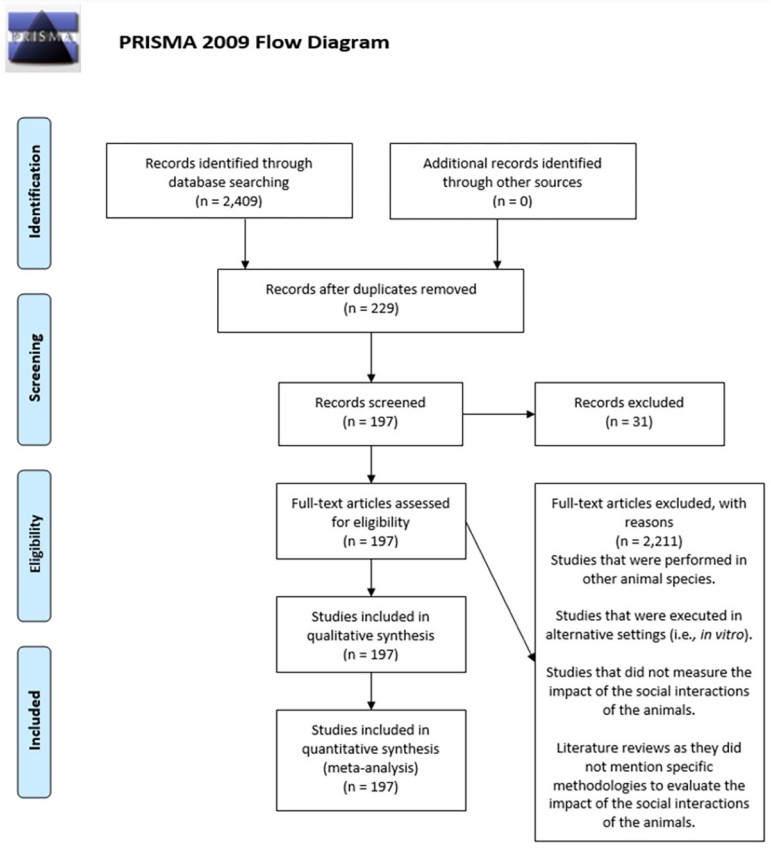

**Fig 4. PRISMA flow diagram for social stress systematic search results.**

Weight gain was the most used indicator for animal performance among the manuscripts reviewed (51 manuscripts, 96.22%), while only 7 manuscripts (3.53%) analyzed the pigs' vocalizations to detect social stress.

## Immune-related stress

A systematic search of studies looking at immune-related stress through November 24, 2020, ultimately yielded 535 relevant manuscripts. A total of 33,727 results were obtained from the initial search, which included immunological challenges, such as vaccine tests, sanitization challenges, microorganism exposure, and any other factors that could affect the immune status of the animals. After excluding studies not relevant to pig production, the number of articles was reduced to 4,309. After further excluding duplicates and other studies that did not fit the search profile, such as those involving animal species other than pigs (such as guinea pigs), or *in vitro* or genomic studies, a total of 535 studies were included for review. The complete list of references used in the systematic literature review for the immune-related model is presented in the S9 Table in S1 File. The manuscripts were classified into separate categories using 4 animal-based indicators: blood chemistry (452 manuscripts, 82.63%), physiological activity (341 manuscripts, 62.34%), animal performance (228 manuscripts, 41.68%), and behavior (82

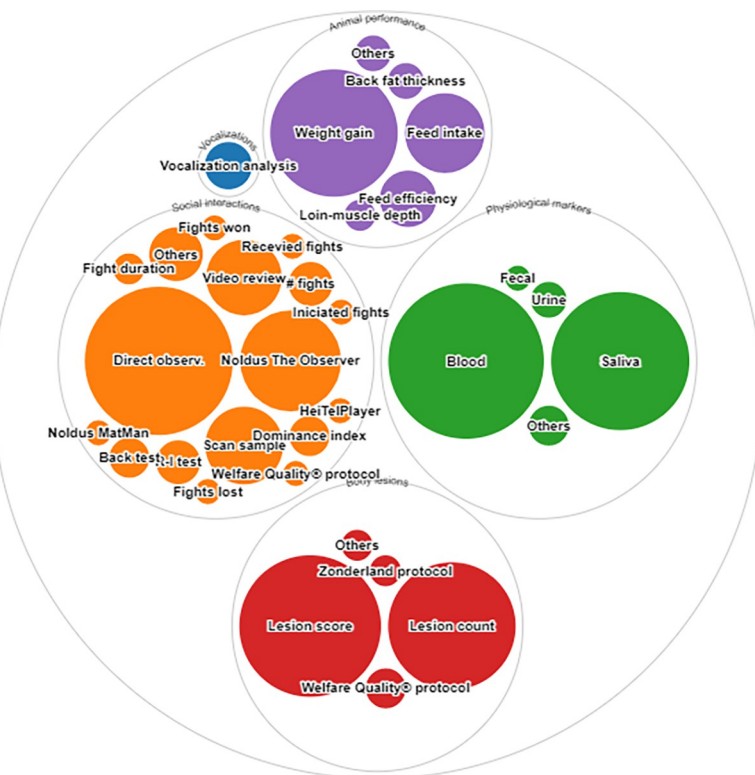

**Fig 5. Methods to measure social stress included animal performance, animal vocalizations, physiological markers, cortisol levels, lesions, and social interactions.** Methods pooled in the "Others" domain are presented in the S5 Table in S1 File. (Source: rawgraphs.io).

manuscripts, 14.99%). The physiological activity category included other indicators related to immune-related stress, such as viral load, organ integrity, body temperature, respiration rate, and swelling, among others (Fig 7).

After reviewing the articles, 82 indicators were identified for the evaluation of the impact of immune-related stress in pigs under either research or production conditions (Fig 8). A list of these indicators is available in the S10 Table in S1 File. According to the initial classification, 6 indicators were identified for the evaluation of animal performance, 6 for behavior, and 49 indicators related to blood analysis. The severity of diarrhea was evaluated using 2 physiological indicators, while 26 measures were used to assess the impact of the immunologic challenges.

Blood chemistry parameters, such as increases in the concentration of cytokine, immuno-globulin, cortisol, and acute-phase proteins, as well as hematology assessments, were the most frequent indicators for evaluating immune-related stress (452 manuscripts, 82.63%). More-over, increased interleukin levels were the most widely reported biomarker in this category, either in serological concentrations, or less frequently, through gene expression levels in differ-ent tissues (398 manuscripts, 88.05%), which we opted to include with the blood chemistry group to avoid confusion. The decision to separate the blood chemistry from the physiological markers was made based on the variety of markers measured in the blood.

Body temperature was the most frequent indicator among the physiological measurements (194 manuscripts, 56.7%). Other relevant indicators were organ integrity (153 manuscripts, 44.73%), respiratory rate (69 manuscripts, 20.17%), intensity of diarrhea (64 manuscripts, 28.36%), and viral load (55 manuscripts, 16.08%).

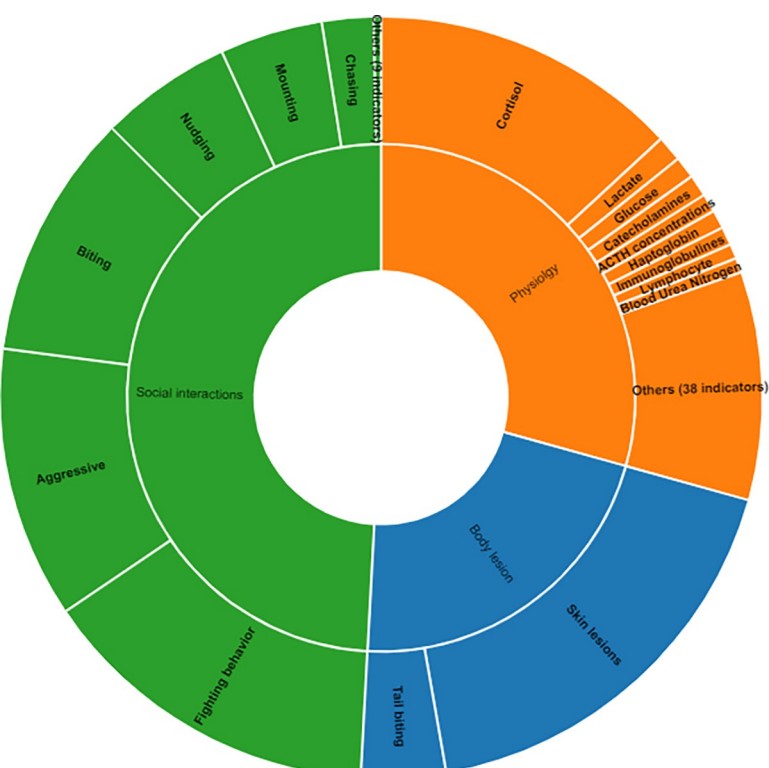

**Fig 6. Physiological markers, lesion assessment, and social behavior indicators found in the literature regarding the social stress model pulled from the databases.** Completed lists of the physiological markers, body lesion markers, and social behavior indicators are presented in the S6-S8 Tables in S1 File. Figure generated through rawgraphs.io.

Animal performance was primarily reported using average daily weight gain, which was described in 187 studies (82.1%), followed by feed intake (165 manuscripts, 72.36%), and feed conversion rate (78 manuscripts, 34.21%).

The use of behavioral indicators was minimal for the evaluation of immune-related stress challenges, only being mentioned in 82 manuscripts (14.99%). Depression at the activity level, mentioned in 59 manuscripts (71.1%), was the most popular behavioral indicator.

### Immune challenges

Immune-related stress has been studied through several microbial challenges (viruses, bacteria, and vaccines, among others). In the present systematic review, 63 immunological challenges were identified. Lipopolysaccharides from *E. coli* were the most widely used (37.2%), followed by porcine reproductive and respiratory syndrome virus (PRRSV) (14.8%) and salmonellosis (6.8%) (Fig 9). A complete list of challenge models found in the present systematic review is presented in the S11 Table in S1 File.

### Discussion

From the results obtained through our systematic literature search and subsequent data processing, the stress-response indicators have been presented based on their relationship with stress. As such, the indicator types are as follows: A) **causal**: indicators that measure the factors that cause stress; B) **biological response**: indicators that measure the physiological response of the organism which help cope with stress; and C) **consequence**: indicators that quantify the

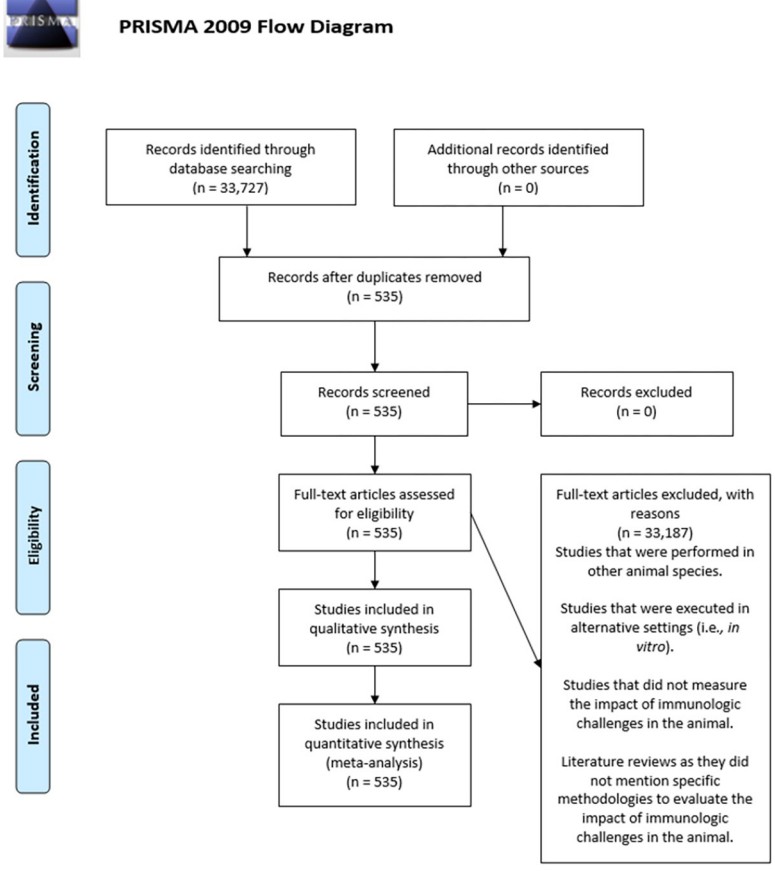

**Fig 7. PRISMA flow diagram for immune-related stress systematic search results.**

productive consequences of the physiological responses to stress. Among these indicator categories, biological response indicators would be those displaying higher sensitivity to stress, as they are aimed at assessing the physiological reactions which take place in an effort to return the organism to homeostasis, followed by the consequence indicators.

Additionally, the indicators were assessed using validity, reliability, and feasibility concepts [26], although it is important to note the limitations of the literature reviewed in the present systematic review. Due to the search terms selected, it is possible that some manuscripts were overlooked. The systematic review never intended to show the complete literature regarding stress in pigs, but to show the most relevant, to provide a clear overview of the methods used to measure the impact of stressors in pigs.

## Thermal stress

Pigs exposed to high temperatures were found to have a decreased voluntary feed intake and an increased respiratory rate, water intake, and peripheral blood flow, which are all aimed at reducing the production of body energy and increasing heat dissipation. Furthermore, the physiological responses to maintain thermal homeostasis require an additional energy expenditure, with a consequently negative impact on productive parameters [67–69]. The primary

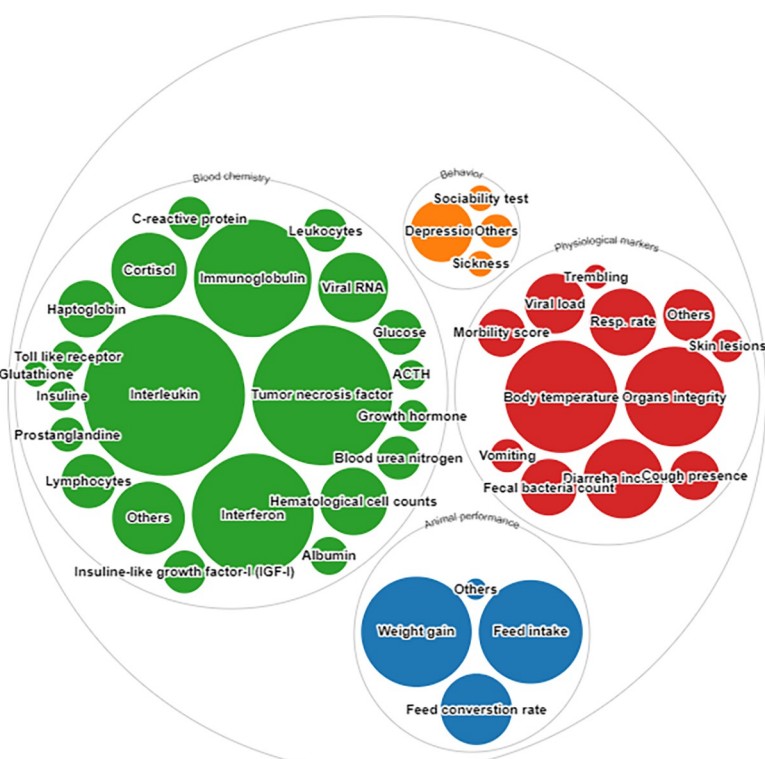

**Fig 8. Animal-based indicators used in the literature from the systematic search for the immune-related stress model.** Methods pooled in the "Others" domain are presented in the S10 Table in S1 File. Figure generated through rawgraphs.io.

strategies pigs use to cope with high temperatures are physiological responses and behavioral reactions. Indicators that are more sensitive to an increase in environmental temperatures are body temperature, respiratory rate, and lying behavior, all of which tend to increase at temperatures higher than 32˚C, when compared to lower temperatures (18, 24, and 28˚C) [2, 5]. Therefore, it is worth focusing on the development of technologies and methods for the monitoring of these indicators, as they may also facilitate the monitoring of thermal stress with non- or minimally invasive methods.

**Causal indicators.** *Environmental indices.* Thermal stress is generated by a combination of environmental factors, such as humidity and ambient temperature, and the anatomical fact that pigs have a decreased perspiration capacity [6, 45, 70–72]. Therefore, environmental indices and mathematical models have been developed to estimate the impact of the rise in environmental temperature on the thermal status of animals [43–46, 64, 65, 73, 74]. In particular, the increase in the indices measured in the studies relative to the thermoneutral conditions of each study were reported in each study. Among the literature reviewed, the temperature-humidity index (THI) has been the most widely used model for predicting pig body temperature, based on environmental temperature and humidity [64]. Environmental indices, such as the black globe temperature index, humidity index, THI, thermal circulation index, and heat load, generate differences in the indices results relative to the animal-based measurements (rectal or skin temperature), due to variations in factors such as wind speed, air pressure, body weight, and feed intake. Therefore, factors other than temperature and humidity need to be considered in environmental indices to improve their accuracy [43, 44, 46, 64, 65].

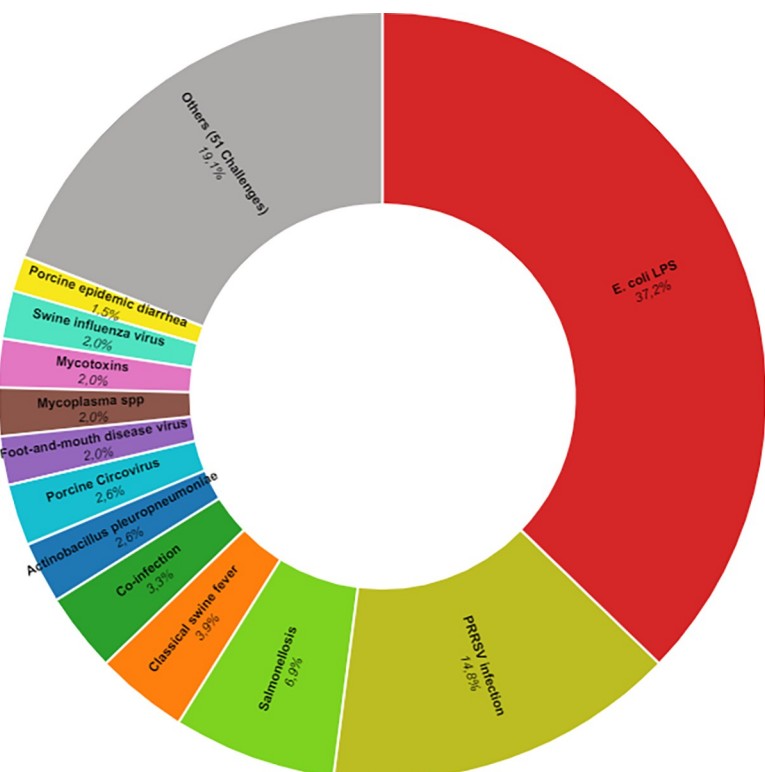

**Fig 9. Immune challenges identified in the literature from the systematic review.** Figure generated through rawgraphs.io.

The validity of environmental indices has been widely reported in the relevant literature [43–46, 64, 65, 73, 74], and the reliability of the indices is well defined as described in the literature. Additionally, environmental indices are a practical technique, as they require only environmental data registration with little subsequent data processing. However, the data processing might require particular attention from personnel to perform the calculations and properly interpret the results.

**Biological response indicators.** *Body temperature.* The determination of body temperature using rectal thermometers has been widely described in the literature related to the surveillance of the thermal status of animals because of its practical execution and accurate measurement (valid, reliable, and feasible) [69, 73, 75–87]. Rectal temperature measurements, however, require handling, which may disturb the animal, potentially altering its thermal status [87, 88]. Additionally, rectal measurements are time-spot measurements, which may limit the acquisition of information for the assessment of animal thermal status [54, 71, 87, 89]. Several studies have tested alternative measurement techniques to validate indicators used to check body temperature, which can monitor thermal status in a continuous fashion with less invasiveness. As such, the frequency of rectal temperature use has increased over the years as the gold standard with which to validate alternative methods [4, 54, 87, 90–93].

*Skin temperature.* A less invasive technique for monitoring the thermal status of pigs is the use of infrared (IR) thermography, which is focused on the detection and increase of IR radiation coming from the skin of pigs subjected to high-temperature environments [50]. The validity of IR skin temperature measurements has been proven by several authors, who have evaluated the accuracy of IR temperature results using rectal temperature as the gold standard

[55, 88, 94–99, 104–108]. The results of these studies have demonstrated the potential use of IR temperatures as a reliable indicator to evaluate the thermal status of animals.

In terms of reliability, skin temperature measurement requires adjustments to improve its accuracy and practicality. One such adjustment is to improve the number of individuals that can be checked at once, as the temperature data of more than one animal will improve certainty about the thermal status of a group of animals. This adjustment, however, also requires individual identification to correlate the measurements taken to a particular animal. Another such adjustment would be to increase the distance needed for an accurate measurement, as IR thermometers usually require the operator to be at a certain distance. Under standard production conditions, however, pigs may constantly move around a pen, which may complicate the measurement process. Additionally, the presence of personnel can alter the behavior of animals by increasing their physical activity (e.g., running around the pens), which would subsequently increase the temperatures of the animals. Another adjustment would be to establish a standard location on the animal where the measurement is taken, as different body parts release more or less thermal radiation than others [54, 74, 94–97]. The feasibility of using IR thermography to measure skin temperature depends on the cost of the technology acquisition. Furthermore, improvements and updates to these technologies will be applied in more animal production units, as long as the prices remain affordable and generate significant economic returns.

In summary, the use of IR radiation measurements has a high potential for the monitoring of body temperature due to its lack of invasiveness and rapid results, which combined with proper data management and interpretation could be used as an early warning system for the thermal status of animals. Nonetheless, there are practical limitations which need to be addressed, such as those mentioned above.

The use of superficial data loggers to record information on skin temperature is gaining attention, as they can obtain continuous information without disturbing the animals [71, 92, 98, 100–102]. Some of these loggers, however, need to be implanted subcutaneously inside the body of the animals or attached as ear tags. Consequently, these methods may induce pain in animals. Furthermore, especially in ear tags, the data logger may be susceptible to damage, loss, or displacement from the implantation area.

*Respiratory rate.* Assessment of increased respiratory frequency provides contactless method for monitoring the thermal state of the animal, as an increased respiratory rate is part of the adaptation strategy of an organism to cope with high temperature conditions to maintain thermoregulation, speeding up cooling through evaporative heat loss [95, 103]. Pigs, in particular, must dissipate heat through perspiration, as they are deficient in sweat glands [72, 78, 95, 103, 104].

The validity of respiratory rate as an indicator has been proven by several authors [45, 70, 83, 105, 106], who have used it as a thermal status indicator (see the Results section). In terms of reliability as an indicator, the current method with which this variable is measured has limitations, as counting the movements of the flank requires a trained observer. Additionally, counting the movements of the flank focuses on just one animal in a group, disregarding information about the status of the other animals within the group. Moreover, it is another time-spot measurement that loses information throughout the day [76, 102, 107–111].

The respiratory rate is a feasible indicator, as it can be performed in most production conditions, although with the previously mentioned limitations (individual animal values and time-spot measurements). To improve efficacy and usefulness, there is a need to develop a continuous and systemic assessment technique for respiratory rate (*refinement*). From the relevant literature reviewed, there are a few innovative alternatives which could potentially optimize the assessment techniques for respiratory rate: one is the use of acoustic recordings to detect the

expiratory and inspiratory sounds of the animals [2], and the other is the use of IR thermography to monitor heart and respiratory rates [112] by measuring the movements of the chest.

*Physiological markers.* The primary physiological markers utilized in detecting the impact of temperature in the animals were the detection of increased glucose concentrations, reductions of the integrity of the ileum, and increases in cortisol concentrations. These physiological markers reflect the general reactions of an organism to the high environmental temperatures mentioned above. Physiological markers are valid, as they have been widely used as a way to understand the functioning of organisms in a variety of studies (nutrition, genetics, reproduction, etc.); however, their use as an indicator of thermal stress depends on the interpretation of the data obtained, and their association with environmental conditions.

The evaluation of physiological indicators involves taking samples from animals, which is associated with some degree of animal handling; and in some cases, such as blood sampling, is also associated with pain [8, 29, 31, 45, 77, 84, 99, 105–107, 111, 113–122]. Additionally, some biomarkers may be altered by handling of the animals and sampling procedures [89]. The reliability of physiological markers as indicators, in general, depends on utilizing an appropriate sampling process, correct sample analysis, and adequate data interpretation. Potential alterations due to handling stress may affect the accuracy of the measurements, and limit the practical applications of the indicator.

Additionally, the analysis of the samples (blood, saliva, feces, hair, tissue, etc.) requires laboratory processing and data interpretation, which might result in extra economic expenses, decreasing its suitability for routine assessment. Moreover, blood or tissue sampling may require the authorization of a bioethics committee, depending on the degree of severity of animal pain and suffering. These limitations may affect the feasibility of using physiological markers as indicators of thermal stress; however, because of their accuracy, physiological markers are still considered a good alternative for detecting thermal stress in pigs.

**Consequence indicators.** *Behavior.* Another indicator of thermal stress is animal behavior, and pigs tend to increase the time spent lying on the ground to maximize heat dissipation from the floor [60, 61, 123–125]. Lying behavior can be evaluated through various methods. One is through the direct observation of lying pigs, either by scan sampling (time-spot sampling), continuous video recording [60, 61, 73, 109, 126] or automatic quantification through machine vision [47, 48, 70, 125–127]. The use of automated algorithms in pigs eliminates the uncertainty of time-spot sampling, and reduces the time delay of the process and data interpretation method, because the behavior of the animal is recorded and analyzed in real time. Machine vision software methods are under constant development, and technology is continuously evolving to handle issues such as individual identification within a group of animals, monitoring during dark hours, and increasing processing speed to generate prompt notifications.

The use of behavior as a thermal stress indicator is limited by its feasibility, as it is a time-consuming task (either through direct observation or video review classification). The refinement of taking measurements through machine vision will improve its application, validity (as more researchers will measure it), and reliability (as computer processing guarantees proper data analysis).

*Animal vocalizations.* Alternative methods utilized to evaluate thermal stress, such as behavior or vocalization analysis, require more technical development, as these techniques have limitations, such as the individualization of animals in the pen, classification data, and technical processing. Sound characteristics (such as intensity, frequency, and tone), however, have accurately provided information about the status of animals, such as pain, thirst, hunger, and extreme temperature [4, 90, 128–133]. As in the behavior indicator section, the development

of technologies relating to vocalizations will improve the feasibility of the application of this indicator, improving its validity and reliability.

Based on the results of the present systematic literature review, the detection of the thermal status of pigs is based on two primary methods: invasive techniques, which aim to measure the inner temperature of the organism (used as the gold standard measurement), and remote techniques, which are designed to assess the thermal radiation of the pigs' bodies (high potential for the refinement of this method, as it involves the application of automated measurement and data interpretation).

The use of less invasive methods is still limited, due to the need for minimal manipulation of the animals (e.g., saliva or urine samples), which can generate confusion, producing a false-positive diagnosis. These methods, however, allow the measurement of physiological indicators that are sensitive to a variety of stressors (e.g., oxidative, immune-related, and social stress), which can facilitate the assessment of the efficiency of strategies to maintain the welfare of pigs and their production level. Therefore, validated measurements and correct data interpretation are necessary to reduce the occurrence of invalid conclusions.

## Social stress

Social stress in pigs can be generated in different ways and at different growth stages during the production cycle. For instance, social stress in pregnant sows can lead to prenatal stress, resulting in piglets with depressed immune capacities [134–138]. Also, it is well known that the pigs are social animals that prefer to live in well-established hierarchical groups [139]; therefore, changes in these social orders may induce aggressive behaviors when accessing resources (food, water, or resting places), which can cause injuries and physiological reactions that decrease animal welfare [139, 140]. This situation is frequent during the weaning process and other regrouping events that frequently occur in standard pig production systems [10, 13, 140–146]. Social stress in pigs can increase the concentration of cortisol, acute phase protein levels, immune activity, and affect several hematological parameters. Additionally, social stress negatively affects pig performance by reducing animal weight gain and affecting the immune development of the offspring [139, 147–149]. As such, physiological and behavioral changes can be used to monitor social stress [139, 140, 150].

**Causal indicators.** *Social interactions*. Behavioral and social interactions can be used to measure the incidence of agonistic behaviors among animals, and the validity of these indicators is reflected in the number of authors who have used them to assess social stress in pigs (see Results section).

An increase in the frequency of specific interactions, such as fighting, biting, nudging, mounting, chasing, and intimidating behavior, has been detected through machine vision and social analysis software packages, such as The Observer series and MatMan from Noldus (Noldus Information Technology, Wageningen, The Netherlands) [13, 151–166], and Interact from Mangold (Mangold International GmbH, Arnstorf, Germany) [167]. The use of software packages to detect social interactions ensures the reliability of the results obtained, as the data computing algorithms have been validated in several studies (see the Results section).

Additionally, researchers still aim to develop and utilize new technologies to detect aggression, such as three-dimensional (3D) cameras [53], motion detection [168], facial expression [169], IR thermography [10], and vocalization analyses [136, 155, 170–172]. These technology-based methods for measuring social interactions are non-invasive, and offer the possibility of monitoring at a group level. Briefly, the sensor captures the data (sounds, images, radiation, etc.) that are sent to an external server and processed in the cloud, and as such, the server generates data records for storage, further processing and interpretation, and electronic

notifications (e.g., warning alerts) to several portable devices (e.g., smartphones). On the other hand, however, animal production sometimes occurs in locations with connectivity deficiencies, as animal production facilities are usually in rural areas, which limits their potential to use internet-based programs [173]. There are several commercial projects that develop sensors and devices to increase the amount of information that can be detected from animals, with the aim of increasing the traceability and sustainability of animal production systems [174]. An alternative to these automated detection methods is the observation of animal interactions through video recording [13, 134, 136, 138, 144, 147, 149, 151, 153, 157, 159, 166, 169, 172, 175–180] or direct observation [35, 140, 165, 170, 171, 181–184]. As mentioned before, however, these methods are limited in the number of individuals that can be assessed at once, as well as in the speed of processing.

Refinement of the measurement of social interactions through technology aims to improve the reliability and feasibility of these indicators. Currently, the identification of behaviors depends on observers to perform the behavior counts, which consequently may reduce the feasibility of these indicator measurements.

*Lesion assessment*. Lesion assessment, through a variety of scores and protocols, can be an indicator of dominant-subordinate relationships, and has been validated in several publications [14, 35, 52, 138, 145, 152, 153, 160, 161, 163, 167, 168, 176, 177, 182, 184–208]. This method assumes that more aggressive animals begin more agonistic encounters, generating higher lesion scores for victimized pen-mates. The measurement method, however, has some limitations that can affect the reliability of the indicator, such as difficulties in detecting other agonistic interactions that do not generate injuries but are still stressors, i.e., nudging or intimidation, and even shorter agonistic and less intense interactions (do not generate observable injuries), which would affect the social dynamics of the individuals involved [142, 186, 209]. Therefore, owing to the variability of this method, it must be performed on large groups of animals to obtain significant conclusions [209]. The measurement protocol of this indicator makes it a feasible indicator for utilization under most production conditions, although body lesion assessment should be included as part of a set of indicators to produce robust and reliable conclusions [26]. For instance, skin lesions complemented by behavioral observations have been used to study the genetic correlation of aggressiveness in pigs, as aggression has been shown to be a heritable component [141, 145, 209–218].

**Biological response indicators.** *Physiological markers*. The primary physiological indicator of stress is an increase in the concentration of cortisol in the blood [12, 137, 140, 147, 160, 178, 179, 183, 184, 219–224] or saliva [13, 35, 36, 51, 149, 152, 159, 175, 181, 187, 190, 206, 210, 222, 225–228]. Additionally, other less invasive measurement techniques such as hair [224] and urine samples [13, 170] have also been used to detect increases in the concentration of cortisol, which is an indicator of the activity of the hypothalamic-pituitary-adrenocortical (HPA) axis, which controls homeostasis in organisms. An increased concentration of cortisol is associated with a reduction in thyroid hormone circulation, a greater risk of ulcers in the stomach, and a higher risk of immune dysfunction, particularly in the intestinal wall [229], increasing the animals' susceptibility to infections and illness. Therefore, cortisol concentration is a relevant biomarker for monitoring pig welfare. Blood cortisol is widely used because of its practical sampling and relatively affordable cost [134, 160, 220, 221]. As previously mentioned, however, blood sampling involves animal handling and discomfort for the animals [230], and requires ethical approval of the sampling protocols. Additionally, blood metabolites, such as cortisol and acute-phase proteins, have a circadian circulation pattern that affects their concentration in the bloodstream during the day [139, 167, 175]. Less invasive sampling methods, on the other hand, such as saliva, feces, and hair, are less stressful to the animals, and sampling can be carried out multiple times with fewer detrimental effects on the integrity of the pig.

Additionally, saliva samples can be used to detect multiple physiological markers [36, 206]. Discord, however, has been observed between salivary and serum cortisol [179, 224], as saliva samples generate a higher variance among the results than cortisol concentrations in the plasma because of the higher variance of this variable and its circadian rhythm [206, 231]. Similarly, hair cortisol measurements have accuracy limitations related to hair factors, such as hair longitude, subject sex, and hair color, which may influence the concentration of cortisol in the sample [224, 232]. Fecal samples have shown a weak relationship with plasma cortisol measurements [233] and the sampling process to obtain fresh and uncontaminated samples directly from the pig rectum might require handling.

Other physiological stress markers, such as increases in glucose [139, 140, 158, 179, 188, 234, 235], acute phase proteins [12, 36, 147, 183, 206, 224], and catecholamines [36, 134, 147, 149, 219, 224], also measured from blood or saliva, have the same limitations as cortisol concentration assessments, such as required handling of the animal, circadian rhythms, sample processing, and data interpretation-related costs. Additionally, the approval of the sampling protocols by a bioethics committee reduces the feasibility of measuring physiological markers.

The validity, reliability, and feasibility of physiological marker indicators are discussed above.

**Consequence indicators.** *Animal vocalizations.* Animal vocalizations might be considered a behavioral indicator, as they are a consequence of environmental stimuli [163]; however, in the present systematic review, vocalizations were considered an indicator category *per se*, as the manuscripts reviewed studied and related the acoustic characteristics of animal vocalizations with particular social, environmental, or health situations [4, 62, 91, 128–130, 132–134]. The use of this indicator is still under development to increase the interpretation capacity of the information captured from the sounds recorded in pig farms, and to improve the identification of specific individuals in a given production group [236, 237]. The validity, reliability, and feasibility of animal vocalizations as welfare indicators have already been mentioned above.

**Animal performance.** Animal performance indicators, such as growth and feed intake, reflect the consequences of stress on the integrity of the animal, which may negatively affect animal production. Nonetheless, these measurements would not be specific enough to elucidate the physiology of the animal under stress conditions by themselves, because performance indicators simply reflect the consequences of stress and not the physiological response to the stress factor. Typically, these indicators are used to complement other indicators [40, 51, 142, 184, 227, 238–240].

The validity of animal performance indicators has been proven, as they are widely used to track the development of animals during the production cycle. The feasibility or ease of practical application makes the results of performance measurements widely reliable.

## Immune-related stress

Immune challenges result in reductions in the physiological functions (maintenance, growth, and reproduction) of the animals, affecting animal performance as well as economic profitability [241, 242]. Common responses to immune challenges include increased pro-inflammatory cytokine activity, such as interleukins (IL-1, IL-2, IL-12, IL-17, IL-18), interferon (IFN-γ), and tumor necrosis factor (TNF-α), an increase in the production of acute phase proteins, and leukocyte proliferation [16]. These responses generate consequences such as fever [16, 87, 243–254], diarrhea [34, 244, 253, 255–259], dyspnea [249, 260–269], spontaneous abortions, and reductions in animal performance parameters (i.e., growth, feed intake, and body conditions) [147, 268, 270–279]. The observation of these consequences or symptoms can provide an idea

of the magnitude of the immunologic challenge, but it cannot provide an ultimate diagnosis of immune function [269, 280]. The primary physiological indicators used to monitor the effects of immune challenges are the activity of cytokines [16, 270, 271–278, 281, 282], immunoglobulin proliferation [268, 279, 283–290], and blood metabolites such as acute-phase proteins, glucose, and cortisol [18, 33, 291–299].

**Causal indicators.** The immune-related stress model does not present causal indicators, as immunological challenges are part of the experimental procedure. Some assessments, however, are aimed at evaluating the viral load of pigs or the presence of viral ribonucleic acid (RNA) in the individual [300–303]. These indicators were reviewed in the physiological indicator methods, as those measures are part of the evaluation of clinical symptoms.

**Biological response indicators.** *Blood chemistry*. Biochemical indicator measurements (cytokines, immunoglobulins, and acute-phase proteins) require animal handling for sampling (blood or tissue). New methods have been developed for the detection of biological markers using less invasive techniques, such as saliva, urine, feces, and nasal swabs [17, 270, 302, 304–315]. These methods have shown the potential to detect metabolites, such as acute-phase proteins, immunoglobulins, and cortisol, in saliva samples [17, 231, 286, 306, 316], immunoglobulins and viruses present in nasal secretions [286, 290, 317–320] and immunoglobulins in fecal samples [321]. An additional advantage of these less invasive methods is the potential habituation of the animal to the sampling process, which would optimize the procedure and improve the welfare of the animals [306]. For example, Almeida [322] collected oral fluid samples from piglets in the lactation area by hanging a cotton rope for one hour. The animals chewed the rope, from which their saliva was extracted. Furthermore, piglets were more prone to contact the sampling device (cotton rope) if their mother was in contact with the object. These less invasive methods, however, are susceptible to higher variation, which require an increased sample size to obtain significant conclusions [231], and some authors have reported results contrary to those found through blood sampling [179, 223]. The validity, reliability, and feasibility of the physiological markers using blood samples as welfare indicators have been mentioned in previous paragraphs.

*Physiological markers*. This category covers methods that measure the physiological impact of immune challenges on animals. In general, the indicators measured were clinical symptoms which presented in the animals when the infection or challenge was administered [323–326]. This indicator category provides information on how and when the stressor affects the physiology of the animals, hampering their welfare and production yield. Additionally, the detection of clinical signs and symptoms requires trained observers [34, 258, 307, 313, 327–329]. The validity, reliability, and feasibility of the physiological markers have been discussed previously in the text.

**Consequence indicators.** *Animal performance*. The physiological reaction chain triggered by an immune challenge manifests in clinical signs that compromise animal performance parameters (feed intake, feed conversion rate, weight gain, body condition score, etc.) [34, 40, 247, 248, 250, 255, 330–345] and increase the incidence of diarrhea [255, 270, 315, 327, 346–352]. As in previous stress models, animal performance might be considered an indicator of other biomarkers. The indicator characteristics (validity, reliability, and feasibility) for the animal performance indicators are described above.

*Behavior* Immunologic challenges reduce the energy available for routine activities, forcing the organism to use this energy to fight infection and maintain homeostasis [241, 242], depressing the level of physical activity [40, 353–356]. Therefore, methods to monitor pig activity remotely and continuously are relevant to production systems. Most of the behavioral analysis packages mentioned in the previous sections, the Observer series and MatMan from Noldus [13, 151, 160–166, 152–159] and Interact from Mangold [167], can guarantee accurate

and fast processing of information regarding the pigs' behavior. Moreover, a variety of sensors and behavioral analysis packages are constantly under development. For instance, the thermographic cameras developed by Flir detect changes in the superficial temperature of specific body parts (udder, hoofs, etc.) or detect fever [357–361]. The validity, reliability, and feasibility of behavior as a welfare indicator have been previously discussed.

The methods used to monitor immune stress require a higher level of refinement than in other stress models, because most of the measurements and assessments require sampling that involves animal handling (blood, tissues, urine, saliva, nasal fluids, etc.). Therefore, methods that allow the determination of several metabolites in a single sample are highly desirable.

## Final considerations (summary)

The primary indicators and methods described in the relevant literature from 2000 to 2020 for each stress model are well known, and we have described the advantages and limitations of these methods for measuring the indicators used to determine the impact of stressful factors on the physiology of animals in porcine production systems. Additionally, future directions that need to be addressed to optimize the techniques and technologies to detect these indicators were described, always leaning towards protocols that are less stressful to the animals while providing more accurate results and improving the general welfare of the pigs.

Among the stress models reviewed, three types of indicators were observed: those that evaluated the cause of the stress, those that measured the biological response to deal with the stressor, and those that assessed the consequences of stress. Therefore, it is important to consider that, to generate a proper measurements and correct diagnoses of the status of an animal, the physiological response enacted to deal with a stressful situation should be measured. As such, researchers and pig producers can determine the magnitude of the impact of the stressor on the physiology of an individual. A summary of the primary indicators and their measurement techniques is presented in Table 2, organized based on the type of indicator.

Additionally, we identified three types of methods that are applicable to all stress models:

(A) invasive methods, such as body temperature through rectal measurements and blood metabolites through blood samples to determine cortisol, acute phase proteins, cytokines, blood urea nitrogen, and glucose; (B) less invasive methods, such as concentration of cortisol, acute phase proteins, and immunoglobulins in saliva; and (C) non-invasive methods, such as animal behavior, skin temperature, and vocalizations.

Invasive methods have a high accuracy, and their performance is relatively affordable and practical. These methods, however, require animal handling, which may alter the levels of the measured metabolites. Additionally, invasive methods might require the approval of a bioethics committee, as these proceedings can generate fear, pain, and distress in animals, which

**Table 2. Primary indicators and measurement techniques of each stress model, presented based on the indicator type.**

| Indicator | Thermal | | Social | | Immune-related | |
|---|---|---|---|---|---|---|
| | **Indicator** | **Method** | **Indicator** | **Method** | **Indicator** | **Method** |
| Causal | • Environmental temp. | • THI | • Social interactions<br>• Lesions | • Direct observ.<br>• Lesion score | • Physiology markers | • Viral Load |
| Biological response | • Body temp.<br>• Skin temp.<br>• Respiratory rate<br>• Physiological markers | • Rectal temp.<br>• NIFT<br>• Flank moves count<br>• Glucose | • Physiological markers | • Cortisol | • Blood chemistry<br>• Physiological markers | • -Interleukin<br>• -Body temp. |
| Consequence | • Behavior<br>• Vocalizations | • Direct observation<br>• Vocal analyses | • Animals perform.<br>• Vocalizations | • Weight gain<br>• Vocal analyses | • Animals perform.<br>• Behavior | • Weight gain<br>• Direct observ. |

might in turn affect the results of the measurements. A limitation of invasive methods is that these are time-spot measurements, which disregard most of the animal's information during a normal day.

Less invasive methods allow for decreased manipulation of the animals (e.g., restraint to collect the sample), and conditioning the individual to get used to the experimental procedure (e.g., offering a cotton swab soaked with a sucrose solution). Additionally, the number of metabolites that can be identified in these samples increases their potential as a more refined method for measuring biological markers. However, circadian fluctuations in metabolites or sampling-handling alterations can affect the variability of measured parameters. These alterations would require an increased sample size to obtain statistically significant conclusions.

Remote methods are under constant development and optimization, as technology and systems are updated regularly. There, the limitations of these methods are the affordability and feasibility of adding technology to the facilities housing the production system. However, accurate measurements and interpretations make it worth investing in improving animal welfare and production yield.

One remarkable point is the emerging need to detect universal indicators that are sensitive to a variety of stressors, which would facilitate the assessment of the physiological status of animals under different challenging conditions. This would allow researchers, producers, and stakeholders involved in pig production to optimize the generation and evaluation of strategies to handle stressors related to pig production to maintain the welfare and production of pigs. Specific parameters, however, are better for identifying specific problems. Additionally, it is important to identify methods that are feasible for utilization in farms to control the physiological conditions of the animals, and to study the relationship of these biomarkers with productive parameters to predict the performance of individuals under standard productive conditions. The use of holistic approaches and the possible complementarity of indicators will provide a more complete landscape of the animals' response to stress factors.

As such, remote observation and machine vision methods are promising alternatives for monitoring animal welfare. Moreover, the development of new technologies and the large-scale production of these devices and data processing packages will likely make this technology more affordable and feasible for application in animal production systems. All these new, less invasive, and remote methods follow the "*refinement*" concept to improve animal welfare, wellness, and measurements. Finally, the utilization of complementary indicators will provide a broader picture of the landscape regarding the physiological status of animals under typical production conditions.

## Conclusion

The present systematic review aimed to review animal-based indicators currently used to monitor the impact of different sources of stress in pigs (thermal, social, and immune-related). The primary indicators and methods used to measure stress have been reported in the relevant literature, and methods that rely on technologies such as artificial vision, sound analysis, and IR technology have been highlighted. These technologies not only improve the accuracy of the measurements, but also reduce animal handling, thereby improving animal welfare. For instance, NIFT can be used to measure skin temperature, record vocalizations to detect stressors, or analyze artificial vision to interpret animal behavior.

Additionally, stress assessments based on a single animal-based indicator may fail to provide a reliable diagnosis of the stress response of an animal. Therefore, researchers have suggested to consider using a set of indicators that cover physiological and behavioral responses to obtain complete stress monitoring.

Finally, stressors are implicit in animal production, and the study of these stressors may lead to more efficient management and nutritional strategies to mitigate the effects of stress on pig production. Current stress models are based on invasive sampling, which might reduce animal welfare. Therefore, studies involving stress models should balance the severity of the invasiveness of the indicators under evaluation with the welfare benefits of the study to obtain reliable results with minimal negative impacts on the welfare of animals.

## Supporting information

**S1 Checklist.**
(DOCX)

**S2 Checklist.**
(DOCX)

**S1 File.**
(XLSX)

## Author Contributions

**Conceptualization:** Raúl David Guevara, Gemma Tedo, Pol Llonch.

**Data curation:** Raúl David Guevara, Gemma Tedo, Pol Llonch.

**Formal analysis:** Raúl David Guevara.

**Investigation:** Raúl David Guevara.

**Methodology:** Raúl David Guevara, Pol Llonch.

**Supervision:** Gemma Tedo, Pol Llonch.

**Validation:** Raúl David Guevara, Gemma Tedo, Pol Llonch.

**Visualization:** Raúl David Guevara.

**Writing – original draft:** Raúl David Guevara.

**Writing – review & editing:** Raúl David Guevara, Jose J. Pastor, Xavier Manteca, Gemma Tedo, Pol Llonch.

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
