## [Decision Letter · Decision Letter 0]

30 Sep 2021

PONE-D-21-22028Systematic review of animal-based indicators to measure thermal, social, and immune-related stress in pigsPLOS ONE

Dear Dr. Guevara Ballesteros,

Thank you for submitting your manuscript to PLOS ONE. After careful consideration, we feel that it has merit but does not fully meet PLOS ONE’s publication criteria as it currently stands. Therefore, we invite you to submit a revised version of the manuscript that addresses the points raised during the review process.

Your manuscript has been reviewed by two experts. Both have very critical views and I agree that, in its present form, the article provides limited interesting information. A deep restructuring should be done and additional information and analysis should be included in order to reach the scientific soundness required for publication in this journal. Please follow carefully all the reviewers comments.

We look forward to receiving your revised manuscript.

Kind regards,

Cristina Óvilo, Ph.D.

Academic Editor

PLOS ONE

Journal Requirements:

Reviewers' comments:

Reviewer's Responses to Questions

**Comments to the Author**

1. Is the manuscript technically sound, and do the data support the conclusions?

Reviewer #1: Partly

Reviewer #2: Partly

2. Has the statistical analysis been performed appropriately and rigorously? 

Reviewer #1: N/A

Reviewer #2: Yes

3. Have the authors made all data underlying the findings in their manuscript fully available?

Reviewer #1: Yes

Reviewer #2: Yes

4. Is the manuscript presented in an intelligible fashion and written in standard English?

Reviewer #1: No

Reviewer #2: Yes

5. Review Comments to the Author

Reviewer #1: Guevara et al. present a review of biomarkers for three stressors in pigs, temperature, social environment, and immune challenges. The review is poorly structured and poorly written. I miss substance I would expect in a good review, novel ideas, novel ways to see things, future trends.

1. There are several similar reviews out there and I do not really see any novelty in this review. I miss a clear aim of this review and an attempt to bring a new aspect.

2. I do not like the classical structure of the review. In this way the review loses flow, is disjointed, and redundant.

3. Another redundancy brings the fact, that the animals have only limited repertoire of stress responses, and most are general. Thus describing each stressor separately brings unnecessary redundancy and the same parameters are presented and discussed repeatedly

4. There is also lack of consistency. In the part on immune challenge we learn that cytokines are being measured also by expression analysis in different tissues. First, this occurs in a paragraph on biomarkers measured in blood. Second, it brings an important aspect that is largely omitted in the review - use of holistic approaches. There is a clear need to use these technologies, and even more so their combination, to develop biomarkers with high diagnostic value. The biomarkers discussed, such as cortisol, are just based on general biological knowledge.

5. The language and also presentation of the review does not make an impressionthat it was written by an expert. For example separation of blood parameters and physiological parameters. These are all physiological parameters, even behavior is a physiological reaction.

6. I miss a more thorough discussion on many aspects. For example the authors mention measuring biomarkers using samples that are minimally invasively collected such as hair etc. Here I miss discussion of the drawback of these approaches not being suitable for longitudinal analyses and not allowing situative assessment of the state of the animal.

I propose to restructure the review. First review the biomarkers. Than review the different stressors first introducing the basic biology behind the response, than the biomarkers being used, and than their strengths and weaknesses, and novel approaches and future trends.

Reviewer #2: The subject is of interest, and it is within the scope of the journal. Certainly we need to enlarge our knowledge on selection and validation of stress markers.

A systematic review a meta-analysis was carried out in a sound manner and a high number of references were found which warranted a nice piece of data set for analysis. Analysis was carried out in a sound manner.

Indicator of stress was differentiated according to thermal, social, and immune-related stress indicators. Authors concentrated their efforts on quantifying the number of variables analyzed out in each case.

I felt a little disappointed to see that not further analysis was carried out that may help to reach a valid conclusion this matter. The fact that many scientists decided to use a certain analysis does not support it validity. Some extra information, such a mean value, sd (or coefficient of variation), % difference vs control groups, correlations among each other or any other information (which I supposed was available in the manuscript) would provide a valid output and certainly a reference to inspiration and discussion.

6. PLOS authors have the option to publish the peer review history of their article (what does this mean?). If published, this will include your full peer review and any attached files.

Reviewer #1: No

Reviewer #2: No

---

## [Author Response · Author response to Decision Letter 0]

15 Nov 2021

Revision Note to manuscript PONE-D-21-22028

We thank the editor and the reviewers for your valuable comments and invested time on reviewing our manuscript. We have taken them into consideration with the aim of fulfilling the information provided, contributing to clarify the objectives to achieve with our systematic review and to better explain the novelty of the document. On the other hand, please, find our answers to the comments posted below. Those are referred as the authors’ response (AR:). 

Editor comments: Your manuscript has been reviewed by two experts. Both have very critical views and I agree that, in its present form, the article provides limited interesting information. A deep restructuring should be done, and additional information and analysis should be included in order to reach the scientific soundness required for publication in this journal. Please follow carefully all the reviewers’ comments.

Authors’ Response (AR): We have amended the manuscript following your comments, which have been seriously taken into consideration to fulfill your requirements. Please, find below our answers trying to better explain our actions to the experts in the revised manuscript.

Reviewer #1: Guevara et al. present a review of biomarkers for three stressors in pigs, temperature, social environment, and immune challenges. The review is poorly structured and poorly written. I miss substance I would expect in a good review, novel ideas, novel ways to see things, future trends.

1. There are several similar reviews out there and I do not really see any novelty in this review. I miss a clear aim of this review and an attempt to bring a new aspect.

AR: Thank you very much for your comment. We have taken it into consideration with the aim of fulfilling the information provided, contributing to clarify the objectives to achieve with our systematic review and to better explain the novelty of the document.

The aim of the review is to provide an exhaustive revision of the main indicators measured in pigs under the most relevant challenges identified in pig production systems (specified in lines 93-99 within the manuscript). This revision includes more than eight hundred manuscripts (876) with relevant information for the subject under research, which are targeting the identification of relevant animal-based indicators according to the main stressor (thermal, social or immune-related) to monitor them to become, potentially, indicators at the farm level. 

According to our knowledge, most of the literature review articles related to stress indicators have been focused on just one stressor (stress model). For instance, Bjerg et al. (2020), and Godyn and Herbut (2018) revised the indicators of thermal stress. Giles et al. (2017) and Adewole et al. (2016) discussed the biomarkers related to immune challenges in pig production systems. Furthermore, some literature reviews have paid their attention basically on the capacity of one methodology to obtain animal-based indicators. For instance, Nasirahmadi et al. 2017 reported the advantages and new trends of behavioral assessment through machine vision, Gutierrez et al. 2014 highlighted the potential saliva proteomic markers for stress assessment, and Cook et al. 2013 reviewed the capacity of infrared thermography to detect changes in body temperature of the pig. In our manuscript we have included all relevant methodologies, indicators and even parameters enabling to evaluate the impact of such stressor in their productivity. We have classified stress indicators in every stress model according to their relationship with the stress response in the animal, therefore, the novelty of our manuscript comes on how those have been classified: Causal indicators, Biological response indicators, and Production consequences indicators. We believe this is a novel approach, aiming to contribute to better monitor the stress response in pig studies, in experimental but also in field conditions. 

References:

- Adewole, D. I., Kim, I. H., and Nyachoti, C. M. (2016). Gut health of pigs: Challenge models and response criteria with a critical analysis of the effectiveness of selected feed additives - A review. Asian-Australasian J. Anim. Sci. 29, 909–924. doi:10.5713/ajas.15.0795.

- Bjerg, B., Brandt, P., Pedersen, P., and Zhang, G. (2020). Sows’ responses to increased heat load – A review. J. Therm. Biol. 94, 102758. doi:10.1016/j.jtherbio.2020.102758.

- Cook, N., and Schaefer, A. (2013). Infrared thermography and disease surveillance. Thermogr. Curr. status Adv. Livest. Anim. Vet. Med., 79–92. Available at: http://www.fabioluzi.it/wordpress/wp-content/uploads/2012/11/THERMOGRAPHY-2013.pdf.

- Giles, T. A., Belkhiri, A., Barrow, P. A., and Foster, N. (2017). Molecular approaches to the diagnosis and monitoring of production diseases in pigs. Res. Vet. Sci. J. 114, 266–272.

- Godyń, D., and Herbut, P. (2018). Applications of continuous body temperature measurements in pigs – a review. Ann. Warsaw Univ. Life Sci. - SGGW - Anim. Sci. 56, 209–220. doi:10.22630/aas.2017.56.2.22.

- Gutierrez, A., Ceron, J., Fuentes-Rubio, M., Tecles, F., and Beeley, J. (2014). A Proteomic Approach to Porcine Saliva. Curr. Protein Pept. Sci. 15, 56–63. doi:10.2174/1389203715666140221115704.

2. I do not like the classical structure of the review. In this way the review loses flow, is disjointed, and redundant.

AR: The structure of the review in this manuscript is firstly trying to classify the most widely used (not necessarily relevant) indicators in each stress model that have been published so far, taking into consideration that the stress models are quite different among them and at the same time considering that several stressors may interact or be present in filed conditions. The organization of indicators by stress model aims to clarify the specific responses of the animal to each stress factor, which elucidates that an indicator may provide different information of an animal depending on the stress factor. For example, a reduction of the animal physical activity level can be observed when a pig is exposed to thermal load (Olivera et al., 2020) and during immune challenges (Brückmann et al., 2020). However, during a social challenge, such as weaning or litter mixes, a higher activity level may reflect the aggressiveness of pigs, which is directly associated to the stress response (Colson et al., 2006).

The review pretends to be an informative document for researchers, and eventually technicians, before selecting the most appropriate animal-based stress indicators, or even to help on selecting the most appropriate indicators to monitor animal welfare at a farm level. Therefore, gathering all indicators from each stress model, and discussing the pros and cons may assist the reader in selecting the most up to date and relevant indicators according to the context of research and evaluation.

Overall, we believe that the current structure may fulfill the objective of this review which aims to become a reference tool for animal scientists but also pig production stakeholders when looking for indicators to monitor stress.

- Brückmann, R., Tuchscherer, M., Tuchscherer, A., Gimsa, U., and Kanitz, E. (2020). Early-life maternal deprivation predicts stronger sickness behaviour and reduced immune responses to acute endotoxaemia in a pig model. Int. J. Mol. Sci. 21, 1–22. doi:10.3390/ijms21155212.

- Colson, V., Orgeur, P., Courboulay, V., Dantec, S., Foury, A., and Mormède, P. (2006). Grouping piglets by sex at weaning reduces aggressive behaviour. Appl. Anim. Behav. Sci. 97, 152–171. doi:10.1016/j.applanim.2005.07.006.

- Oliveira, A. C. F., González, J., Asmar, S. E., Batllori, N. P., Vera, I. Y., Valencia, U. R., et al. (2020). The effect of feeder system and diet on welfare, performance and meat quality, of growing-finishing Iberian × Duroc pigs under high environmental temperatures. Livest. Sci. 234, 103972. doi:10.1016/j.livsci.2020.103972.

3. Another redundancy brings the fact that the animals have only limited repertoire of stress responses, and most are general. Thus, describing each stressor separately brings unnecessary redundancy and the same parameters are presented and discussed repeatedly. 

AR: We agree with the reviewer that the repertoire of stress responses is limited. The structure of the review intends to remark the different reactions of an animal to each stressor and how an indicator may provide different information depending on the stressor. The current structure aims to show the possible complementation among these indicators. For instance, the information of body temperature can be complemented with behavioral parameters to identify if a rise in body temperature is due to a high environmental temperature or generated by an immune challenge that produces a febrile response. The idea of highlighting the complementarity of the indicators is present in the revised manuscript and we believe that this aspect is relevant to be considered, as no single indicator may provide a definite picture of the stress response towards single or even different stressors.

4. There is also lack of consistency. In the part on immune challenge we learn that cytokines are being measured also by expression analysis in different tissues. First, this occurs in a paragraph on biomarkers measured in blood. Second, it brings an important aspect that is largely omitted in the review - use of holistic approaches. There is a clear need to use these technologies, and even more so their combination, to develop biomarkers with high diagnostic value. The biomarkers discussed, such as cortisol, are just based on general biological knowledge.

AR: We agree with the reviewer that this part might bring some confusion. Please, find in the revised manuscript (Lines 308-310) clarifications concerning the subject of cytokines’ measurements in tissues.

5. The language and also presentation of the review does not make an impression that it was written by an expert. For example separation of blood parameters and physiological parameters. These are all physiological parameters, even behavior is a physiological reaction.

AR: We agree with the reviewer that blood chemistry indicators belong to the physiological markers group. However, we decided to split them in a more specific group due to the huge variety and nature of blood markers that can be assessed. For example, the intention was to highlight the relevance of cytokines as the main immune-related indicators without minimizing the other physiological parameters, such as the detection of clinical symptoms (body temperature, respiration rate, diarrhea incidence, among others). We appreciate the reviewer's comment pointing out the need for an explanation to clarify this aspect, please, find it in the revised manuscript clarifications concerning the subject (Lines 310-312). 

6. I miss a more thorough discussion on many aspects. For example the authors mention measuring biomarkers using samples that are minimally invasively collected such as hair etc. Here I miss discussion of the drawback of these approaches not being suitable for longitudinal analyses and not allowing situate assessment of the state of the animal.

AR: Please, find in the revised manuscript amended lines adding information related to the consideration of the cortisol measurement in hair samples within the discussion referred to the social stress model (Lines 606 – 608). Additionally, we have added information concerning the drawback of fecal cortisol measurement that we think may complement the information about the less invasive methodologies (Lines 608 – 610). 

I propose to restructure the review. First review the biomarkers. Than review the different stressors first introducing the basic biology behind the response, than the biomarkers being used, and then their strengths and weaknesses, and novel approaches and future trends.

AR: Thank you for this valuable suggestion. We have seriously taken this approach into consideration. We considered that we should keep the structure of the review although implementing changes to improve its scope, willing to become a reference tool for researchers when evaluating any pig stress model. Please, find two main aspects that we have discussed within the reasoning: First, we believe that keeping the current structure based on the responses by stress model (thermal, social and immune-related), we could introduce the specific biological responses of the animal to each stressor, and the animal-based indicators to each stress factor. Even if stress responses are limited, they are different in magnitude and intensity according to each stressor. Thus, reviewing all the biomarkers together would depict the idea that all the stressors produce the same biological responses, which we believe is not correct. Secondly, presenting the stress biomarkers altogether without the differentiation of the stress source would minimize the significance of some indicators due to the amount of differences in the literature reviewed in each stress model (Thermal: 144 articles, Social: 197 manuscripts, and Immune-related: 535 papers). Therefore, the immune-related indicators would eclipse the importance of stress markers for the thermal and social stress models, particularly the ones with a lesser proportion in the revision. For example, the novel approaches, such as the animal vocalization analyses that are relevant for social stress, would be diminished and overlooked. 

We appreciate very much the opportunity given by the reviewer to better explain and defend the purpose of the manuscript under revision. 

Reviewer #2: The subject is of interest, and it is within the scope of the journal. Certainly we need to enlarge our knowledge on selection and validation of stress markers.

A systematic review a meta-analysis was carried out in a sound manner and a high number of references were found which warranted a nice piece of data set for analysis. Analysis was carried out in a sound manner. Indicator of stress was differentiated according to thermal, social, and immune-related stress indicators. Authors concentrated their efforts on quantifying the number of variables analyzed out in each case. I felt a little disappointed to see that not further analysis was carried out that may help to reach a valid conclusion this matter. The fact that many scientists decided to use a certain analysis does not support it validity. Some extra information, such a mean value, sd (or coefficient of variation), % difference vs control groups, correlations among each other or any other information (which I supposed was available in the manuscript) would provide a valid output and certainly a reference to inspiration and discussion.

AR: We acknowledge the positive comments by the reviewer. We agree with the reviewer's suggestion on the valuable information that may be provided by adding a meta-analysis. However, we attempted a robust systematic review with clear endpoints to focus on identifying all the references available in the different stress models and to see the indicators that have been used. Altogether, the large variability among studies did not enable us to run a single meta-analysis, so we have decided to keep it as a systematic review. Please, see some additional comments that were considered to take such decision:

1. A meta-analysis would have restricted the number of studies considered because many papers would have been discarded due to divergent experimental conditions. We preferred to gather an extensive dataset, spotting the maximum number of references, and focus on the capacity of a measurement to generate reliable results at any kind of experimental setting. Hence, we present the number of times that a certain indicator has been used no matter the experimental conditions.

2. We decided to mention the indicator direction change (it increase, decreases, or equals the value relative to the control) to help the reader to understand what is detecting the measurement methodology, and what is representing the indicator value. These changes are presented in the discussion section (see the following lines):

- 373 – 375; 390; 404 – 405; and 438 for the thermal stress model.

- 538; 570 – 571; 585; and at 631 – 633 for the social stress model.

- 665 for the immune-related stress model.

 We hope that experts take into consideration our amendments, clarifications and reasoning for further evaluations of the manuscript.

As mentioned, we thank all valuable comments and suggestions addressed by reviewers to improve the manuscript and their dedication and willingness to refine the quality of our systematic review.

---

## [Editor Report · Decision Letter 1]

25 Jan 2022

PONE-D-21-22028R1Systematic review of animal-based indicators to measure thermal, social, and immune-related stress in pigsPLOS ONE

Dear Dr. Guevara Ballesteros,

Thank you for submitting your manuscript to PLOS ONE. After careful consideration, we feel that it has merit but does not fully meet PLOS ONE’s publication criteria as it currently stands. Therefore, we invite you to submit a revised version of the manuscript that addresses the points raised during the review process.

We look forward to receiving your revised manuscript.

Kind regards,

Cristina Óvilo, Ph.D.

Academic Editor

PLOS ONE

Additional Editor Comments:

Thanks for the effort made to explain and justify the structure and rationale for presentation of data.

Authors claim that the manuscript has been amended according to our comments, but in fact no restructuring was done at all, and no new information was included, with the revised version being practically the same as the original one, which is disappointing. Besides the argumentation for the reviewers, authors should make an effort to improve the paper, (which is the goal of the review process) for instance by highlighting the novel aspects of the work (possibly in the objectives and in the conclusion section which should be newly written, see last sentence below) or adding some data as suggested by reviewer 2

Besides, the manuscript still needs a very deep revision of English language and writing before it can be accepted for publication. Some examples (lines correspond to the submitted manuscript with track changes) are listed below, but this is not a comprehensive list, please revise the whole text

Lines 41-44: this sentence lacks a verb?

Lines 70-71: this sentence is weird, please rewrite

Line 82: change "de" to "the"

Line 98: peer-reviewed

Line 114: counted

Line 118: methodologies for measurement of indicators

Line 124: impact of the stress ON the animals

Lines 187-188: something is lacking

Line 196: physiological markers

Lines 208-209: blood sampling is not a methodology for glucose measurement, it is just the source

Lines 199-225: I don't see the need of subheadings for the presentation of these results, same for the other types of stress model

Line 240: The term Physiology" can not be used as adjective

Line 242: "assessed with the assessment" please rewrite

Line 245: effect ON

Line 246: change "includes" to "include"

Line 263: with 162 papers reporting this type of indicator?... please phrase a little bit....

Line 266: change "in" to "for"

Line 269: more reiterated methodology

Line 298: delete "assess"

Line 304: which supplementary file?

Lines 306-307: delete "."

Line 312: raisings?

Line 317: "Cytokines` gene expression measurement could be fit into the physiological markers category, but it was decided to keep it at the blood chemistry group to avoid confusion. Separation of the blood chemistry from the physiological markers was decided due to the variety of markers that are measured at blood".

Measurement of citokine....

Change "could be fit" to "could fit"

Line 346: causes the stress

Lines 353-357: repetition with intro

Line 383: Particularly, the increase of the index relative to animals in thermo neutral conditions. Which index, you haven't mentioned a particular one, do you mean each index?

Line 386: what is black globe temperature?

Lines 394-396: generation of notifications? what do you mean?

Line 414: has been

Line 419: delete "it"

Line 430: will depend

Line 432: production units / generate

Line 435: because of their LACK OF invasiveness

Line 439: On the other hand

Line 443: induce pain to the animal

Line 452: change stratus to status

Line 471: different kinds of studies

Line 488: are still considered

Line 512: change thirsty to thirst

Line 532: these social orders

Line 538: social stress affects

Line 539: reduces and risks

Line 544: these indicators

Line 546: "The frequency increase " Change to increase in frequency of... has been detected

Line 571: these indicators

Line 572: depends

Line 578: "This methodology assumes that animals more aggressive will begin more agonistic encounters" Change to more aggressive animals

Line 614: "hair cortisol measurements has" change to "have"

Line 621: than cortisol

Line 640: consequences on the integrity

Line 648: amply?? do you mean widely?

Line 669: delete "organism of the"

Line 673: "Biochemical indicators rises measurements" does not make sense. Do you mean rised measurements of biochemical indicators?

Line 696: delete" for its detection"

Line 722: require

Line 723: involve

Line 779: "The use of holistic approaches and the possible complementary of indicators ..." complementary of indicators does not make sense. Do you mean complementarity?

The text included in the conclusions section does not fit in that heading, but is in fact a joint discussion for the different types of stress and their indicators. This part should me moved to discussion under a proper heading, and a new conclusion section should be written
---

## [Author Response · Author response to Decision Letter 1]

3 Mar 2022

We thank the editor for your valuable comments and invested time on reviewing our manuscript. We have taken them into consideration with the aim of fulfilling the information provided, contributing to clarify the conclusion section and the grammar of the manuscript. On the other hand, please, find our answers to the comments posted below. Those are referred as the authors’ response (AR:).

Editor’s comments: Authors claim that the manuscript has been amended according to our comments, but in fact no restructuring was done at all, and no new information was included, with the revised version being practically the same as the original one, which is disappointing. Besides the argumentation for the reviewers, authors should make an effort to improve the paper, (which is the goal of the review process) for instance by highlighting the novel aspects of the work (possibly in the objectives and in the conclusion section which should be newly written, see last sentence below) or adding some data as suggested by reviewer 2.

Authors’ Response (AR): We acknowledge the comments of the editor about the reviewers’ comments on the structure of the literature review. We tried to justify the review structure used in the manuscript, with detailed examples of the novelty of the structure relative to the reviewees available on the literature. Additionally, we have made the suggested changes to increase the clarity of the manuscript. For instance, we have clarified cytokines method of measurement in tissues (Lines 305-309), we added information regarding hair cortisol measurement and the possible weakness of fecal samples as less- invasive method (Lines 603 -607), and we have included information about the indicator direction change (it increase, decreases, or equals the value relative to the control). We believe that this may help the reader understanding what is being actually being measured, and what is representing the indicator value (Lines 363-365, 383-286, 395-397, 431-434, 452-454, and 477-478 for the thermal stress model; Lines 534-538, 566-568, 582-584, and 627-628 for the social stress model; Lines 663-666, 679-680, 687-690, 694-696 for the immune-related stress model). We think that the comments provided by the reviewers and the editor have helped to generate a fine piece of literature that accomplishes to show the trends of measurement methods and indicators to monitor stress in pig production.

Besides, the manuscript still needs a very deep revision of English language and writing before it can be accepted for publication. Some examples (lines correspond to the submitted manuscript with track changes) are listed below, but this is not a comprehensive list, please revise the whole text

Lines 41-44: this sentence lacks a verb?

Lines 70-71: this sentence is weird, please rewrite

Line 82: change "de" to "the"

Line 98: peer-reviewed

Line 114: counted

Line 118: methodologies for measurement of indicators

Line 124: impact of the stress ON the animals

Lines 187-188: something is lacking

Line 196: physiological markers

Lines 208-209: blood sampling is not a methodology for glucose measurement, it is just the source

Lines 199-225: I don't see the need of subheadings for the presentation of these results, same for the other types of stress model

Line 240: The term Physiology" can not be used as adjective

Line 242: "assessed with the assessment" please rewrite

Line 245: effect ON

Line 246: change "includes" to "include"

Line 263: with 162 papers reporting this type of indicator?... please phrase a little bit....

Line 266: change "in" to "for"

Line 269: more reiterated methodology

Line 298: delete "assess"

Line 304: which supplementary file?

Lines 306-307: delete "."

Line 312: raisings?

Line 317: "Cytokines` gene expression measurement could be fit into the physiological markers category, but it was decided to keep it at the blood chemistry group to avoid confusion. Separation of the blood chemistry from the physiological markers was decided due to the variety of markers that are measured at blood".

Measurement of citokine....

Change "could be fit" to "could fit"

Line 346: causes the stress

Lines 353-357: repetition with intro

Line 383: Particularly, the increase of the index relative to animals in thermo neutral conditions. Which index, you haven't mentioned a particular one, do you mean each index?

Line 386: what is black globe temperature?

Lines 394-396: generation of notifications? what do you mean?

Line 414: has been

Line 419: delete "it"

Line 430: will depend

Line 432: production units / generate

Line 435: because of their LACK OF invasiveness

Line 439: On the other hand

Line 443: induce pain to the animal

Line 452: change stratus to status

Line 471: different kinds of studies

Line 488: are still considered

Line 512: change thirsty to thirst

Line 532: these social orders

Line 538: social stress affects

Line 539: reduces and risks

Line 544: these indicators

Line 546: "The frequency increase " Change to increase in frequency of... has been detected

Line 571: these indicators

Line 572: depends

Line 578: "This methodology assumes that animals more aggressive will begin more agonistic encounters" Change to more aggressive animals

Line 614: "hair cortisol measurements has" change to "have"

Line 621: than cortisol

Line 640: consequences on the integrity

Line 648: amply?? do you mean widely?

Line 669: delete "organism of the"

Line 673: "Biochemical indicators rises measurements" does not make sense. Do you mean rised measurements of biochemical indicators?

Line 696: delete" for its detection"

Line 722: require

Line 723: involve

Line 779: "The use of holistic approaches and the possible complementary of indicators ..." complementary of indicators does not make sense. Do you mean complementarity?

AR: We appreciate the time and commitment of the editor to review the language grammar. We made all changes suggested by reviewers’ comments. Additionally, a language review was performed by a professional scientific literature editing service. The language review track changes file and the language review certificate are uploaded with the manuscript. 

The text included in the conclusions section does not fit in that heading but is in fact a joint discussion for the different types of stress and their indicators. This part should me moved to discussion under a proper heading, and a new conclusion section should be written

AR: We thank the editor for your valuable comments about the discussion and conclusions sections. We have taken them into consideration and the amendments have been done to the text aiming to clarify the conclusion section of the manuscript. The new conclusion section can be found at Lines 773 – 791.

We hope that experts take into consideration our amendments, clarifications and reasoning for further evaluations of the manuscript.

As mentioned, we thank all valuable comments and suggestions addressed by reviewers to improve the manuscript and their dedication and willingness to refine the quality of our systematic review.

---

## [Editor Report · Decision Letter 2]

23 Mar 2022

Systematic review of animal-based indicators to measure thermal, social, and immune-related stress in pigs

PONE-D-21-22028R2

Dear Dr. Guevara Ballesteros,

We’re pleased to inform you that your manuscript has been judged scientifically suitable for publication and will be formally accepted for publication once it meets all outstanding technical requirements.

Kind regards,

Cristina Óvilo, Ph.D.

Academic Editor

PLOS ONE
---

## [Editor Report · Acceptance letter]

19 Apr 2022

PONE-D-21-22028R2 

Systematic review of animal-based indicators to measure thermal, social, and immune-related stress in pigs 

Dear Dr. Guevara:

I'm pleased to inform you that your manuscript has been deemed suitable for publication in PLOS ONE. Congratulations! Your manuscript is now with our production department. 

Kind regards, 

on behalf of

Dr. Cristina Óvilo 

Academic Editor

PLOS ONE